# Improving Spatial Disaggregation of Crop Yield by Incorporating Machine Learning with Multisource Data: A Case Study of Chinese Maize Yield

Shuo Chen [1,2,3,4], Weihang Liu [1,2,3,4], Puyu Feng [5], Tao Ye [1,2,3,4,*], Yuchi Ma [6] and Zhou Zhang [6]

[1] State Key Laboratory of Earth Surface Processes and Resource Ecology (ESPRE), Beijing Normal University, Beijing 100875, China; shuochen@mail.bnu.edu.cn (S.C.); 202131051098@mail.bnu.edu.cn (W.L.)

[2] Key Laboratory of Environmental Change and Natural Disasters, Ministry of Education, Beijing Normal University, Beijing 100875, China

[3] Academy of Disaster Reduction and Emergency Management, Ministry of Emergency Management and Ministry of Education, Beijing 100875, China

[4] Faculty of Geographical Science, Beijing Normal University, Beijing 100875, China

[5] College of Land Science and Technology, China Agricultural University, Beijing 100193, China; 2020141@cau.edu.cn

[6] Department of Biological Systems Engineering, University of Wisconsin-Madison, Madison, WI 53706, USA; ma286@wisc.edu (Y.M.); zzhang347@wisc.edu (Z.Z.)

* Correspondence: yetao@bnu.edu.cn

**Abstract:** Spatially explicit crop yield datasets with continuous long-term series are essential for understanding the spatiotemporal variation of crop yield and the impact of climate change on it. There are several spatial disaggregation methods to generate gridded yield maps, but these either use an oversimplified approach with only a couple of ancillary data or an overly complex approach with limited flexibility and scalability. This study developed a spatial disaggregation method using improved spatial weights generated from machine learning. When applied to Chinese maize yield, extreme gradient boosting (XGB) derived the best prediction results, with a cross-validation coefficient of determination ($R^2$) of 0.81 at the municipal level. The disaggregated yield at 1 km grids could explain 54% of the variance of the county-level statistical yield, which is superior to the existing gridded maize yield dataset in China. At the site level, the disaggregated yields also showed much better agreement with observations than the existing gridded maize yield dataset. This lightweight method is promising for generating spatially explicit crop yield datasets with finer resolution and higher accuracy, and for providing necessary information for maize production risk assessment in China under climate change.

**Keywords:** maize yield; spatial disaggregation; machine learning; multisource data

## 1. Introduction

Global climate change poses a great threat to the production, access, use, and stability of the food system [1–3]; thus, future food security is at stake, combined with increasing food demand. Efforts have been devoted to developing a better understanding of the spatial and temporal variations in crop yield, and its response to climate change, for mitigation and adaptation purposes. Previous studies have mostly been based on crop yield as recorded by station observations or by the administrative-level census. However, site-based data are limited in representing large regions [4–7], while statistical yields provide only regional averages, failing to present yield variations caused by intraregional differences in environmental conditions [8–11]. In response, spatially explicit crop yield datasets with continuous long-term series can overcome the shortcomings of station or administrative yields, enabling a better understanding of yield gaps, crop responses to environmental

stress, and the adaptation of cropping systems [12,13], while also offering the necessary information for varying management inputs and insurance or land markets [14,15].

The impact of climate change on China's food security is of great significance to the entire world [16]. China is the largest producer of rice and wheat and the second-largest producer of maize in the world [17,18], helping to feed 22% of the global population on only 7% of the global cultivated land, which has suffered the most from extreme climate events [19,20]. However, China has vast cultivated land areas across regions with diverse climates, soil conditions, and management schemes, resulting in a huge spatial heterogeneity in crop yields [21,22]. Only by considering as many sources of different environmental information as possible can we better reproduce the spatial distribution of crop yield in China.

To date, spatial disaggregation is the most widely used method for generating gridded crop yields with a long time-series at a large scale. Administrative-region census yields are disaggregated with gridded weights using either a simple spatial disaggregation method or a complex hybridization method [23,24]. The simple spatial disaggregation method allocates administrative polygon yields uniformly to all grids within it [25–27]. Consequently, the resultant dataset contains no spatial heterogeneity information. The complex hybridization method optimizes ancillary data (e.g., population density, crop suitability, and irrigation) to create plausible gridded weights. However, previous research has usually used a single parameter—such as net primary production (NPP) or population density—as the weight, but has seldom considered environmental factors comprehensively [28,29]. The spatial production allocation model (SPAM) was the first spatial disaggregation method to use multisource data. However, the SPAM model is too complex for researchers to reproduce or alter when emerging data become available [23]. Therefore, a lightweight and robust model that can flexibly consolidate multisource data to produce more accurate weights is still needed.

Machine learning is a promising tool to improve existing spatial disaggregation methods. Machine learning can flexibly and reliably integrate multisource data, and has achieved impressive success in yield prediction. For example, it has explained more than 75% of spatiotemporal variations in the yield of maize, wheat, rice, and soybeans—not only in China, but also worldwide [30–33]. Several studies have shown its effectiveness in generating gridded yields within small study areas during short periods [30,34–36]. However, machine learning has not yet been applied to spatial disaggregation, and its ability to reproduce the gridded maize yield in China on a large scale with a long time-series remains unknown.

The main objective of this study was to develop a spatial disaggregation method to improve the accuracy of gridded yield by using maize in mainland China as an example. Compared with previous yield spatial disaggregation methods, the proposed method uses machine learning to fuse multisource data and generate a direct weight to disaggregate statistical yields. We aimed to answer the following two research questions: (1) What is the contribution of machine learning algorithms and multisource data to accuracy improvements of maize yield estimation? (2) How much better are the gridded yields generated by the proposed method, compared to existing gridded yield datasets?

## 2. Materials and Methods

### 2.1. Data and Variables

#### 2.1.1. Data

The statistical maize yield at the municipal and county levels from 2000 to 2016 was obtained from statistical yearbooks, with a temporary lack of information in Hong Kong, Macao, and Taiwan. Yields were computed as the ratio of production to the sown area if the yield was not directly provided. Site-level yields were acquired from 99 agricultural meteorological stations (Figure S1). The municipal-level statistical yield was used for modeling, while the county-level and site-level yields were used for cross-scale validation. Global gridded yield maps, including EarthStat, MapSPAM, and GDHY, were obtained for comparison. EarthStat was generated by a simple disaggregation method, and does not

describe intra-county information [27]. MapSPAM was generated by a cross-entropy model driven by spatial constraints, including cropland extent, crop potential suitability, etc. [37]. The GDHY map was generated based on the weights derived from the product of NPP and the harvest index [12,29,38].

Multisource data for climate, remote sensing, soil, and management were considered to disaggregate historical maize yield. Two climate datasets were used, including a 1 km monthly temperature and precipitation dataset for China from 1901 to 2017 [39], as well as TerraClimate [40]. Peng's dataset was spatially downscaled from the 30′ Climatic Research Unit (CRU) time-series dataset and the climatology dataset of WorldClim, and evaluated using observations collected by 496 weather stations across China, showing higher accuracy than the CRU [39]. TerraClimate was produced by a climatically aided interpolation method and water balance model, and it performed with higher accuracy and greater spatial realism than other coarser-resolution gridded datasets [40]. Vegetation index and land surface temperature were collected from MOD13A2, MYD11A2, and MOD11A2, because MODIS imagery has an overall higher quality calibration and longer records [35]. Solar-induced chlorophyll fluorescence was obtained from a global spatially contiguous solar-induced fluorescence dataset (CSIF) [41]. For management data, crop calendars were obtained from agroclimatic stations. The Nutrient Application for Major Crops dataset was used to provide information about the total applied amounts of fertilizer [42]. For soil data, the physical and chemical characteristics of topsoil (0–30 cm) were obtained from the Harmonized World Soil Database (HWSD) [43].

### 2.1.2. Variables

Seven climate variables, including mean near-surface air temperature, maximum near-surface air temperature, minimum near-surface air temperature, precipitation, downward shortwave flux at the surface, vapor pressure deficit, and the Palmer drought severity index, were considered in estimating maize yield. These variables have been included in explaining maize yield for a long time [44–48], because the growth and yield formation of maize are significantly affected by climatic conditions [49,50]. The normalized difference vegetation index and enhanced vegetation index were used to reflect the biomass accumulation of maize, and previous research showed that they were closely associated with maize yield [48,51–55]. Solar-induced chlorophyll fluorescence was used to capture the impact of drought and heat stress on maize, which was significantly related to maize yield in the USA [56–58]. We also took daytime land surface temperature and nighttime land surface temperature into account, because previous studies showed that the correlation coefficients between maize yield and land surface temperature were greater than 0.5, and their relative importance was sometimes higher than air temperature [54,59–61].

Management measures and soil conditions played an important role in crop growth and yield formation and were considered to be the main factors in restricting maize yield and its stability in China [46,62]. Nine promising soil predictors that could better capture the spatial differences in maize yield were considered, including the cation-exchange capacity of soil, cation-exchange capacity of clay, clay fraction, organic carbon, pH, sand fraction, silt fraction, and soil moisture [53,63,64]. Three fertilizer variables—containing the total nitrogen application, phosphorus application, and potassium application—were determined to represent the management measures. The year was added as the temporal variable to explain the increasing trend in the maize yields from 2000 to 2016.

### 2.2. Methods

This study employed a four-step procedure to spatially disaggregate the statistical yield from the municipal level to 1 km grids (Figure 1): (1) pre-processing of data; (2) fitting of empirical models for yield prediction; (3) spatial disaggregation of statistical yields; and (4) accuracy assessment and comparison.

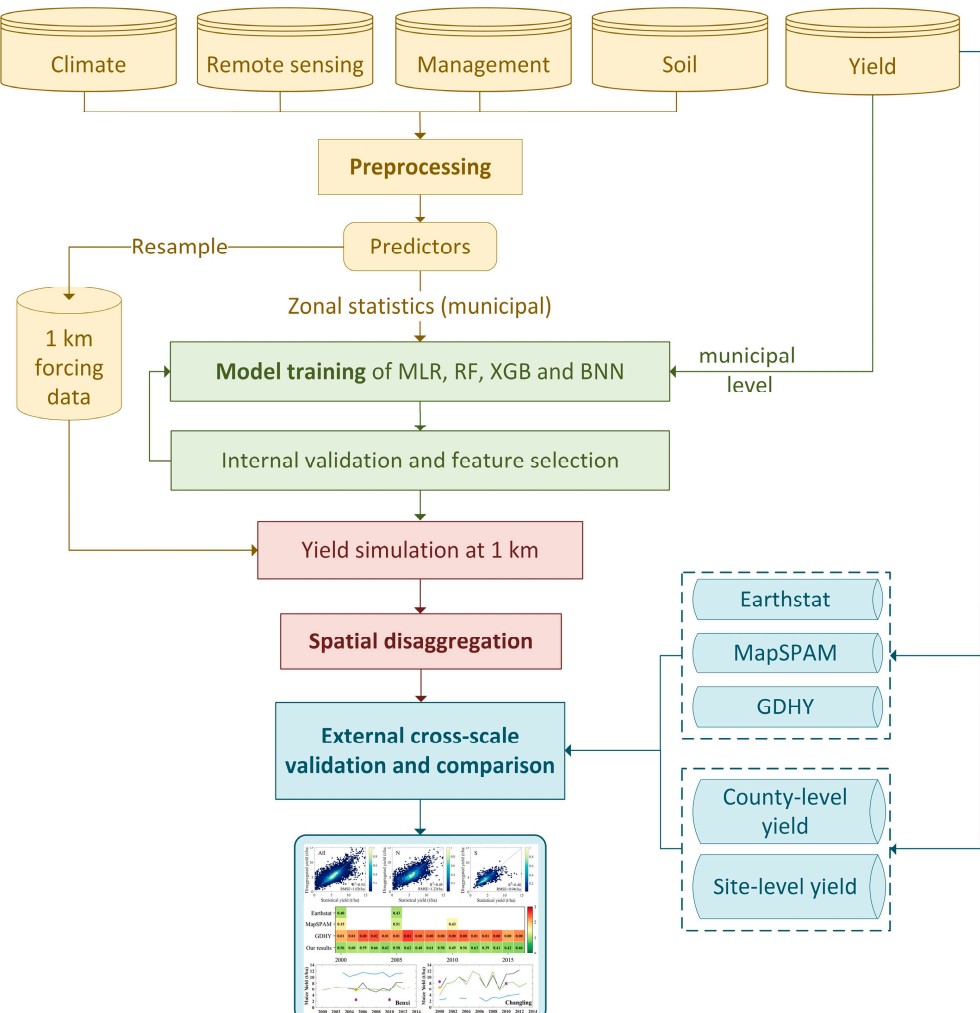

**Figure 1.** The framework of improving spatial disaggregation of maize yield by incorporating multisource data with machine learning.

### 2.2.1. Preprocessing

Crop calendars recorded by stations were interpolated into grids using the nearest-neighbor interpolation method. The planting month was defined as month 1 to harmonize the order of months from planting, based on the planting and harvesting month of the crop calendars. The dynamic data whose temporal resolution was higher than monthly was composited into monthly and growing-seasonal predictors according to specific rules (Table 1). For example, the maximum normalized difference vegetation index value was calculated monthly and by growing season, named "NDVI$_m$" and "NDVI$_{gs}$", respectively; the data whose original temporal resolution was monthly were only composited to growing-seasonal predictors. For example, the average value of monthly mean near-surface air temperature during the growing season was calculated and named "TMP$_{gs}$." Lastly, the SPAM maize harvest area was used as a mask to extract climate, remote sensing, management, and soil predictors. After that, predictors with various spatial resolutions were resampled to 1 km and then aggregated according to the municipal boundary for model training.

**Table 1.** Detailed information about the data collected in this work.

| Dataset | Data Source | Original Resolution | Predictors | Description |
|---|---|---|---|---|
| Yield data | https://data.cnki.net/Yearbook/Navi?type=type&code=A (accessed on 8 May 2022) | Annual, 2000–2016 county and municipal-level | — | Yield |
| | China Meteorological Administration (https://data.cma.cn/, accessed on 8 May 2022) | Annual, 2000–2013 site level | — | Yield |
| | EarthStat [27] | 5-year average, 2000 and 2005 10 km | — | Yield |
| | MapSPAM [37,65,66] | 3-year average, 2000, 2005 and 2010 10 km | — | Yield |
| | GDHY [29] | Annual, 2000–2016 0.5° | — | Yield |
| Climate data | a 1-km monthly temperature and precipitation dataset for China from 1901 to 2017 [39] | Monthly, 2000–2016 1 km | $Tmp_m$ $Tmp_{gs}$ | Mean near-surface air temperature (TMP) for month $m$ of the growing season ("$gs$") |
| | | | $Tmx_m$ $Tmx_{gs}$ | Maximum near-surface air temperature |
| | | | $Tmn_m$ $Tmn_{gs}$ | Minimum near-surface air temperature |
| | | | $PRE_m$ $PRE_{gs}$ | Total precipitation |
| | TerraClimate [40] | Monthly, 2000–2016 4 km | $VPD_m$ $VPD_{gs}$ | Mean vapor pressure deficit |
| | | | $SRAD_m$ $SRAD_{gs}$ | Mean downward shortwave flux at the surface |
| | | | $PDSI_m$ $PDSI_{gs}$ | Mean Palmer drought severity index |
| Remote sensing data | MYD11A2 and MOD11A2 | 8 day, 2000–2016 1 km | $LSTD_m$ $LSTD_{gs}$ | Maximum daytime land surface temperature |
| | | | $LSTN_m$ $LSTN_{gs}$ | Minimum nighttime land surface temperature |

**Table 1.** *Cont.*

| Dataset | Data Source | Original Resolution | Predictors | Description |
|---|---|---|---|---|
| | | 16 day, 2000–2016<br>1 km | $NDVI_m$<br>$NDVI_{gs}$ | Maximum normalized difference vegetation index |
| | MOD13A2 | 16 day, 2000–2016<br>1 km | $EVI_m$<br>$EVI_{gs}$ | Maximum enhanced vegetation index |
| | CSIF<br>[41] | 16 day, 2000–2016<br>0.05° | $SIF_m$<br>$SIF_{gs}$ | Maximum solar-induced chlorophyll fluorescence |
| Management data | Fertilization [42] | Static, 2000<br>0.083° | NAT | Nitrogen application total |
| | | | PAT | Phosphorus application total |
| | | | KAT | Potassium application total |
| | — | Annual, 2000–2016 | year | Prediction year |
| | Crop calendar (https://data.cma.cn/, accessed on 8 May 2022) | Annual, 2010–2013<br>site level | — | Planting and harvest months |
| Soil data | MapSPAM [37,65,66] | 3-year average, 2000, 2005, and 2010<br>10 km | — | Harvest area |
| | HWSD [43] | Static, 2007<br>1 km | CEC_SOIL | Cation exchange capacity of soil |
| | | | CEC_CLAY | Cation exchange capacity of clay |
| | | | CLAY | Clay fraction |
| | | | OC | Percentage organic carbon |
| | | | pH | PH |
| | | | SAND | Sand fraction |
| | | | SILT | Silt fraction |
| | TerraClimate [40] | Monthly, 2000–2016<br>4 km | $SM_m$<br>$SM_{gs}$ | Mean soil moisture |

Notes: "—": not applicable; the subscript *m* stands for monthly values, while the subscript *gs* stands for growing-season values.

### 2.2.2. Model Training

Multivariate linear regression (MLR), random forest (RF), extreme gradient boosting (XGB), and Bayesian neural network (BNN) models were used to fit all of the municipal-year maize yield samples. MLR was conducted in this study to compare the above machine learning algorithms [67].

$$\ln Y = a_1 v_1 + \ldots + a_n v_n + \varepsilon \tag{1}$$

where $Y$ is the municipal yield, $a_1, \ldots, a_n$ are the parameters to be fit, $v_1, \ldots, v_n$ are the predictors, and $\varepsilon$ is the error term.

RF and XGB are both state-of-the-art tree-based ensemble methods that employ a collection of learning algorithms to achieve better predictive power than could be gained from any of these algorithms alone [68]. RF is a combination of tree predictors, such that each tree depends on the values of a random vector sampled independently and with the same distribution for all trees in the forest (Figure S3). The final predicted yield of RF is derived by averaging the predicted yields from all of the individual regression trees. The generalization error for forests converges to a limit as the number of trees in the forest becomes large. The generalization error of a forest of tree classifiers depends on the strength of the individual trees in the forest and the correlation between them [69]. XGB sequentially builds the model—adding a tree each time in XGB is fitting the residual of the previous prediction with a new function [70]. Unlike RF, each tree in XGB is fitted on a modified version of the original training dataset (Figure S4). XGB generalizes boosting methods by allowing the minimization of an arbitrary differentiable loss function; thus, it has a highly efficient realization of the gradient boosting and showed the best performance in recent machine learning challenges [71].

A BNN is a type of artificial neural network (ANN) that models the synapses in a biological brain, where the signal transmits from one neuron to another. The "signal" in ANNs is a real number, and the output of each neuron is calculated as the sum of its input according to specific nonlinear functions. The traditional ANNs need a considerable number of training samples to prevent the model from overfitting. However, BNNs introduce probability distributions over the weights in the neurons and are less prone to overfitting [72]. In this study, the BNN had an input layer with 256 neurons, two hidden layers with 128 neurons, and an output layer with two fully connected hidden layers, with 64 and 32 neurons each, respectively (Figure S5).

The MLR and RF models were trained using the LinearRegression and RandomForestRegressor modules from the package *sklearn*, while the XGB model was trained using *XGBRegressor* from the package *xgboost* in python. The BNN model was developed in TensorFlow 2.0 with Python version 3.7. To compare the performance of different models, the machine learning models were trained with default parameters. The default parameter values of RF and XGB can be found in Table S1, and more details about the settings of the BNN can be found in previous research [72]. To reveal the contribution of multisource data, the four models were driven by nine combinations of input data (Table 2). Seventy-five percent of the complete dataset was randomly selected to train each model, and the remaining data were used to validate the model. The coefficient of determination ($R^2$) and root-mean-square error (RMSE) between statistical and estimated yields were calculated to assess the predictive accuracy, according to Equations (2) and (3), respectively:

$$R^2 = 1 - \frac{\sum\limits_{n=1}^{N} (r_n - f_n)^2}{\sum\limits_{n=1}^{N} (r_n - \bar{r}_n)^2}, \tag{2}$$

$$RMSE = \sqrt{\frac{1}{N} \sum\limits_{n=1}^{N} (r_n - f_n)^2}. \tag{3}$$

where $N$ is the number of samples, $r_n$ and $f_n$ refer to the municipal statistical yield and estimated maize yield, respectively, and $\bar{r}_n$ is the averages of $r_n$.

**Table 2.** The combinations of predictors.

| Abbreviation | Predictors |
|:---:|:---:|
| c | Only climate predictors |
| r | Only remote sensing predictors |
| c + m | Climate and management predictors |
| r + m | Remote sensing and management predictors |
| c + s | Climate and soil predictors |
| r + s | Remote sensing and soil predictors |
| c + m + s | Climate, management, and soil predictors |
| r + m + s | Remote sensing, management, and soil predictors |
| c + r + m + s | Climate, remote sensing, management, and soil predictors |

### 2.2.3. Spatial Disaggregation

After selecting the best-performing model, a leave-one-out method was adopted to identify the outliers, with estimated yields biased by more than three standard deviations relative to the observed yields [73]. The best-performing model was run using the default parameter values to identify the outliers (Table S1). The outlier elimination process can eliminate certain unrepresentative samples, which may introduce errors into the models [73–75]. Then, predictors with importance of over 1% were retained, according to feature importance in the *scikit-learn* module. Feature importance assesses how much each predictor decreases the weighted impurity. This procedure can effectively reduce the number of predictors, and further prevent models from overfitting the highly dimensional training data [76]. Lastly, the grid search was used to tune hyperparameters by trying all of the possible combinations of hyperparameter settings and comparing the out-of-bag data errors of the models with different combinations of parameters.

The gridded predictors at 1 km resolution were used to force the fine-tuned models and generate the gridded estimated yield of 2000–2016. Then, the estimated yield was used to compute the gridded distribution weight $w_{cit}$:

$$w_{cit} = \frac{y_{cit}^{sim}}{\frac{1}{I}\sum_{i=1}^{I} y_{cit}^{sim}}, \tag{4}$$

where $y_{cit}^{sim}$ represents the estimated yield of the grid $i$ located in the municipality $c$ in the year $t$, and $I$ is the number of grids located in this city. The weight $w_{cit}$ was then used to disaggregate the related statistical yield $Y_{ct}^{yb}$ to pixels. The disaggregated yield was produced by using the following equation:

$$y_{cit} = Y_{ct}^{yb} \cdot w_{cit} \tag{5}$$

### 2.2.4. External Cross-Scale Validation

A common method to assess the accuracy of disaggregated yield is aggregating it to a finer spatial resolution and comparing it with statistical yield [12,37,77]. Such a procedure was conducted at the county level in this study. In addition, disaggregated yields were also validated using yield time-series from agrometeorological stations, which were superior to the cross-validation at the county level, but the number of agrometeorological stations was limited. For comparison purposes, the validation procedure included data from three existing global datasets: EarthStat, MapSPAM, and GDHY. We harmonized the temporal resolution to match different datasets to enable such a comparison. Our results were calculated to 5-year average yields in two periods—2000–2002 and 2003–2007 to compare

with EarthStat. Three-year average yields of our results in three periods—2000–2001, 2004–2006, and 2009–2011—were calculated to match MapSPAM. For GDHY, the annual yields enable a year-to-year comparison. After that, the harmonized yield maps were aggregated to the county level, and comparisons were made with the county-level statistical yields. The $R^2$ and RMSE between statistical and disaggregated yields were calculated to assess the accuracy of gridded yield maps, according to Equations (2) and (3).

## 3. Results

### 3.1. Model Training Results

#### 3.1.1. The Contribution of Machine Learning Approaches and Multisource Data

The model training statistics with various combinations of predictors and models are summarized in Figure 2. The machine learning models always outperformed the MLR. The XGB model showed better prediction skills than RF, and was comparable with the BNN. XGB showed a stronger ability than RF to integrate multisource data; the $R^2$ of XGB increased as more predictors were adopted, while the $R^2$ of RF reached the maximum (0.67) after integrating three types of predictors. The BNN outperformed XGB when it was driven by the combination of climate, management, and soil predictors, but XGB generally showed slight outperformance over the BNN when driven by other combinations.

Adding predictors tended to improve the models' performance by increasing $R^2$ and reducing RMSE. If only one group of predictors was used, the climate predictors outperformed remote sensing predictors. Taking XGB as an example, the $R^2$ of the model with climate predictors was 0.47–0.15 greater than that of the model with remote sensing predictors only. When adding a second group of predictors, soil data improved model performance more than management data. The models generally had the highest accuracy forced by the full combination of multisource data (c + r + m + s). As the XGB model with a full list of predictors performed the best, we only present those results in the following sections. After parameter tuning, the $R^2$ between the estimated yields of XGB and the reported yields increased from 0.74 to 0.81 (Figure S6).

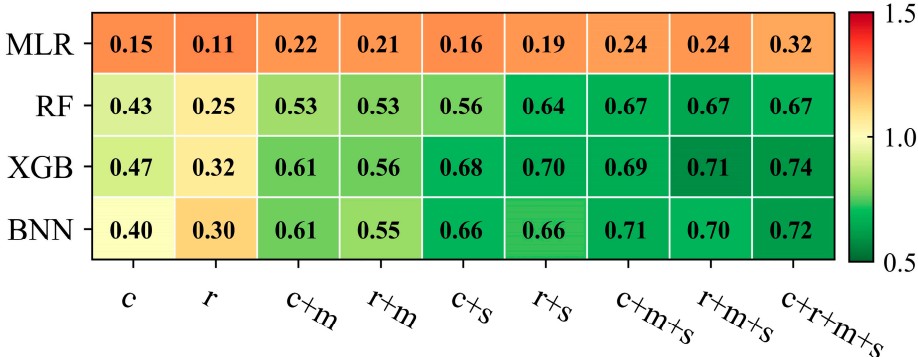

**Figure 2.** Model performance of four models with different combinations of predictors (the numbers represent the coefficient of determination (predicted $R^2$), while the shading colors represent the root-mean-square error (unit: t/ha); c: climate predictors, r: remote sensing predictors; m: management predictors; s: soil predictors).

#### 3.1.2. Feature Importance

The relative importance in maize yield prediction of climate, soil, remote-sensing, and management predictors was 39%, 29%, 17%, and 15%, respectively (Figure 3). There was a total of 23 predictors with relative importance greater than 1%. Among them, there were 10 climate predictors. Growing-season total precipitation (PREgs; 13%), the growing-season mean downward shortwave flux at the surface (SRADgs; 5.6%), and the fourth-month mean vapor pressure deficit (VPD4; 5%) were the top three climate predictors. Five soil predictors entered the final model, and the top three were the cation-exchange capacity of clay (CEC_CLAY), silt fraction (SILT), and percentage of organic carbon (OC),

whose relative importance accounted for 12%, 5.9%, and 4.1%, respectively. There were four remote sensing predictors. Among them, the fourth-month maximum solar-induced chlorophyll fluorescence (SIF4) and third-month maximum enhanced vegetation index (EVI3) were the top two predictors. They made a relatively equal contribution to maize yield prediction, at 6.9% and 5.1%, respectively. There were three management predictors, among which the nitrogen application total (NAT) made the greatest relative contribution, at 7.9%.

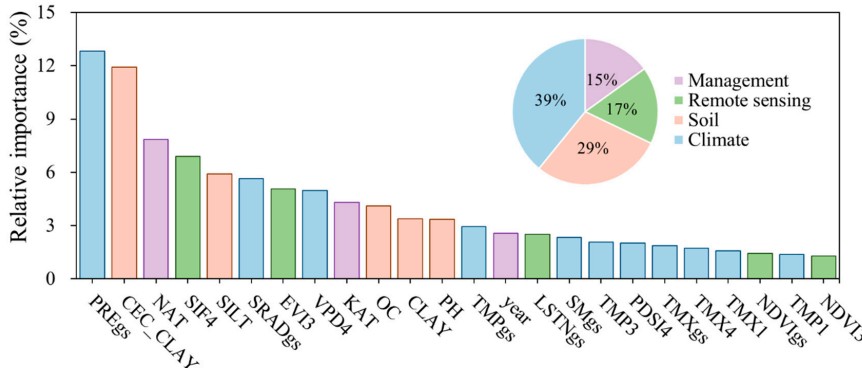

**Figure 3.** The relative importance of selected predictors in XGB.

### 3.2. Validation

### 3.2.1. Cross-Validation at the County Level

When cross-validated with the county-level statistical yield (Figure 4), our disaggregated yield can explain around 54% of the county-level yield variation in mainland China, with an RMSE of 1.02 t/ha. The gridded yield performed best in the Northern China maize zone and the Huang-Huai-Hai maize zone—two maize-planting zones. For regions with sparsely sown areas, it was more challenging to derive reasonable cross-validation scores, i.e., the northwestern maize zone.

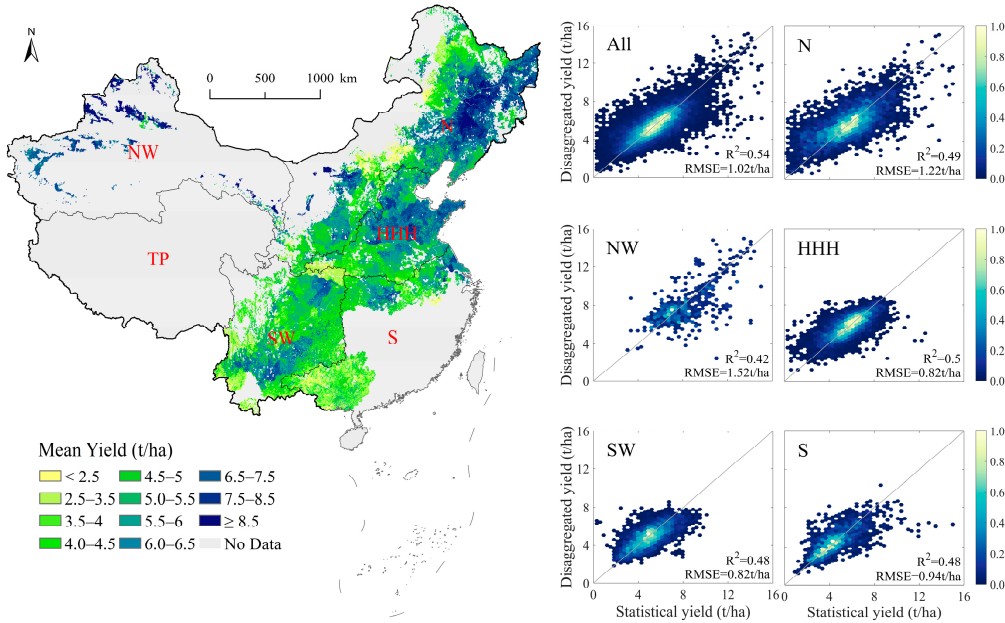

**Figure 4.** Seventeen-year average maize yield distribution (**left**) and hexbin (**right**) for our results and county-level statistical yield from 2000 to 2016 (north (N) spring maize zone; Huang-Huai-Hai (HHH) summer maize zone; southwest (SW) maize zone; south (S) maize zone; northwest (NW) maize zone). The gradient color from blue to yellow (from 0 to 1) represents the density of points. For example, 0 means the lowest density, while 1 means the highest density.

Our method largely improved upon the accuracy of the gridded yield from the existing gridded maize yield datasets (Figure 5). Compared with EarthStat, the RMSE of our results was reduced by 16%, and the $R^2$ increased by around 0.13. Compared with MapSPAM, the RMSE of our results was reduced by 29–41%, and the $R^2$ increased by about 0.17, showing much better consistency with the statistical yields. Notably, there is a cutoff at 10 t/ha in MapSPAM2000, inconsistent with the statistical yields. Compared with GDHY, the improvement of our results was even greater than that of the previous two datasets. GDHY offered continuous annual yield series, but its $R^2$ was rather small (<0.02), and the RMSE was mostly greater than 2 t/ha.

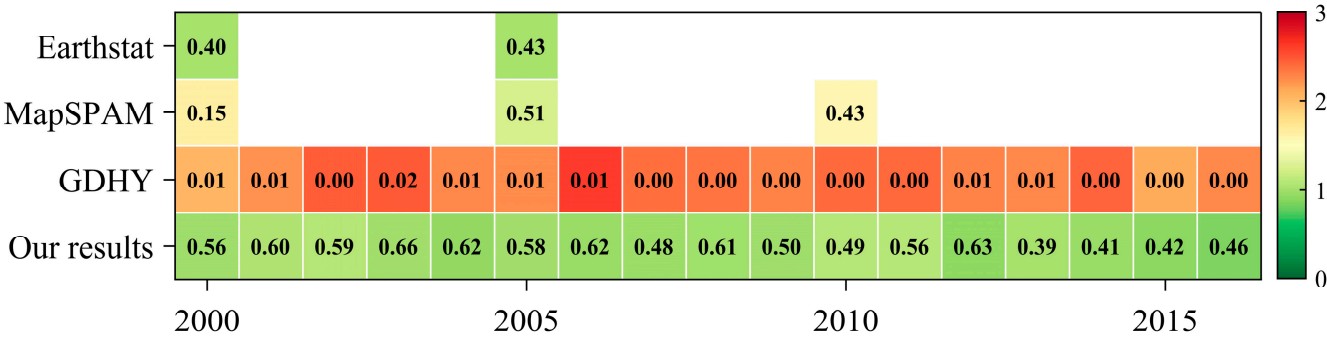

**Figure 5.** Year–county combination comparison between our results and the existing maize yield datasets (the numbers represent the coefficient of determination, and the shading colors represent the root-mean-square error (unit: t/ha)).

### 3.2.2. Cross-Validation at the Site Level

In general, the yield series of our disaggregated results could capture the mean and interannual variation of the station records. Among the 99 sites (Table S2), the average $R^2$ of our results was 0.30, and the average RMSE was 2.39 t/ha, outperforming the other datasets. The only dataset with continuous yield was GDHY, and its average $R^2$ was 0.28, while its average RMSE was 3.17 t/ha. The RMSE of our results was 0.3 t/ha lower than that of EarthStat and 0.81 t/ha lower than that of MapSPAM for specific years. The yield series of the 24 best-documented sites are shown in Figures 6 and S7. At most sites located in the major croplands, the yield series of our results (green lines) were much closer to the observations (black lines) than other yield datasets. However, in the minor corn-growing area, the capability of our method to capture the average and variability of maize yields was similar to that of other datasets, e.g., Qidong station in Jiangsu Province. In addition, there is a discontinuity in our results (green line) for Bachu station in 2009, because there were missing values in the remote sensing images.

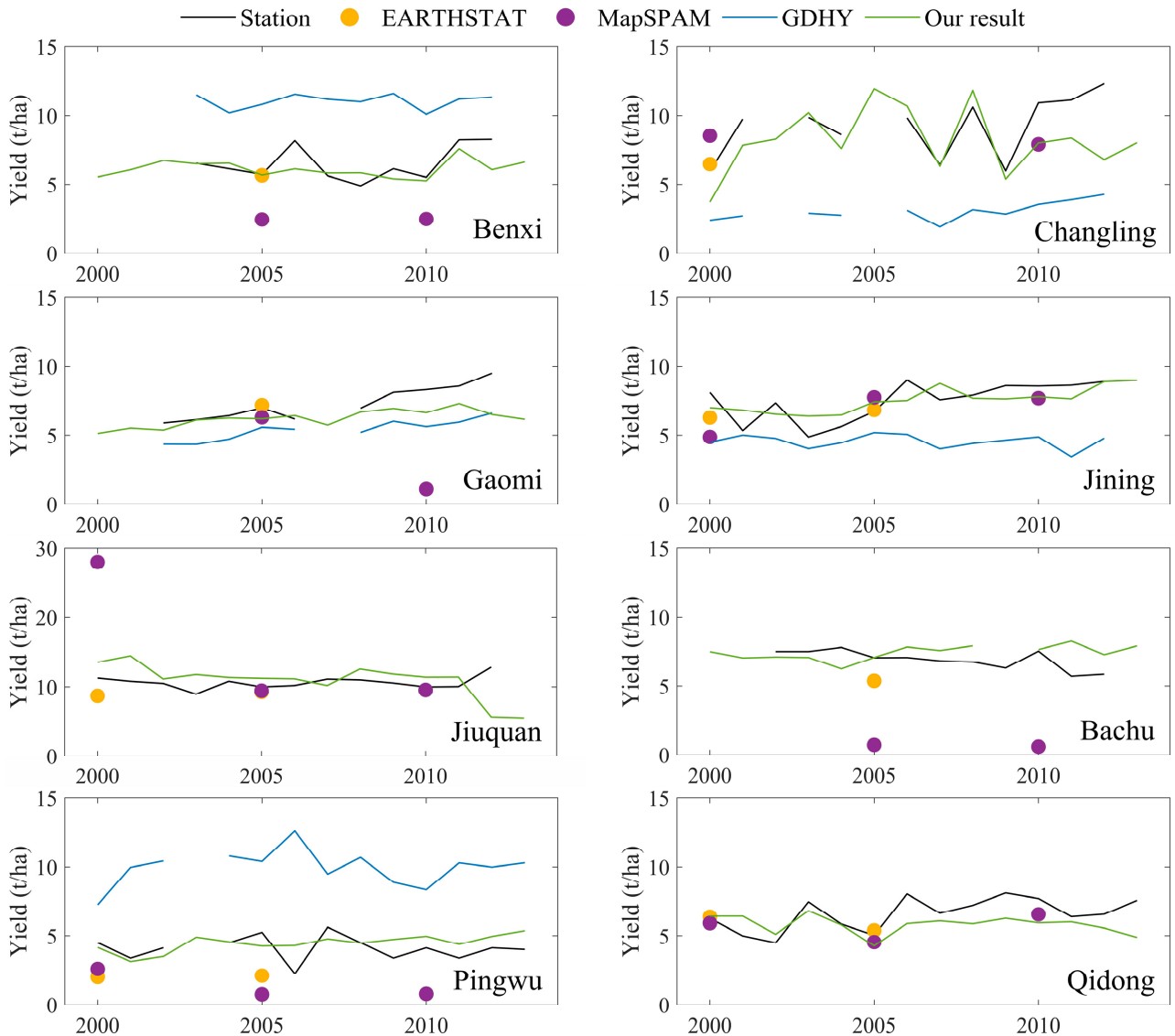

**Figure 6.** Yield time-series in the selected locations for different datasets.

## 4. Discussion

### 4.1. Machine Learning and Multisource Data Improved the Spatial Disaggregation Method

This study took advantage of multisource data and machine learning algorithms in yield prediction, and applied yield prediction results as spatial weights to disaggregate historical maize yield. Our cross-validation results indicated that the proposed method could reasonably generate spatiotemporally continuous gridded yields to large extents. This method benefits from two factors: first, the multisource data that contain rich information about maize growth and yield with various spatial resolutions, and second, the machine learning algorithm that could integrate the complicated relationships between these data.

Compared to our approach, existing spatial disaggregation methods mostly rely on only a few ancillary data, without considering the potential nonlinear and complex internal relationships. MapSPAM is another extreme case that uses a cross-entropy model to integrate multiple sources of information. Nevertheless, the MapSPAM predictors are fixed, and it is hard for researchers to explore the new sources of input information. Our cross-validation results indicate that our method could produce gridded data records slightly better than MapSPAM, and with higher flexibility.

Our method is promising in generating gridded yields at higher resolution because it can flexibly take more predictors with higher temporal and spatial resolution into ac-

count, while the spatial resolutions of existing datasets remain dominant at the 10–55 km level [26,27,37]. Our approach could readily be applied to disaggregate crop yields at various scales and resolutions. Most of the predictors used to generate weights in this study are available in the Google Earth Engine. National or subnational yield statistics can be collected from global agencies, including the Food and Agriculture Organization (FAO) and the World Bank.

Our method could be further improved by incorporating the machine learning model with process-based crop models such as the Agricultural Production Systems Simulator (APSIM) to predict maize yield grown in environments that have yet to be observed [72,78,79]. For example, we can expand the size of the training samples by simulating yield responses under different extreme environments on APSIM to help improve the generalizability of the machine learning model. A similar idea was adopted in the Scalable Crop Yield Mapper (SCYM) to estimate gridded yield in areas with few observational samples. The SCYM uses the pseudo-observations generated by APSIM as response variables and has achieved encouraging success in the USA and Africa [14,80,81].

### 4.2. Models' Performance and Feature Importance in Maize Yield Prediction

In this study, we employed climate, remote sensing, soil, and management data. Our results indicated that their full combination was critical to providing complementary information in yield prediction. Among those four groups of predictors, climate predictors were the most relevant for maize yield in this study, because they provide background weather and external stressor information. However, our results greatly differed for individual climate predictors concerning the relative importance of precipitation and temperature. Growing-seasonal total precipitation (PREgs) was the most important predictor, while the contribution of growing-seasonal average temperature (TMPgs) was much smaller than that of precipitation (PREgs) here. This result is consistent with recent maize research conducted in Germany [82], but opposite conclusions were found in the Midwestern USA [61,72]. This may be because water stress has a stronger influence on maize yield than temperature stress in China, since more than 70% of the covered area affected by agrometeorological disasters was induced by drought or flood [83,84], and the very weak correlations between precipitation and yield in the USA may be the result of better irrigation conditions, larger irrigated areas, and more advanced equipment in the USA.

Soil predictors in our model ranked second in terms of contribution. However, soil predictors were less frequent in previous research, and their importance also remains controversial [53,64,72,85]. For instance, Crane-Droesch (2018) used 39 soil predictors for maize yield prediction, but found them to be the least important among climate, management, and soil predictors. A possible explanation for this disagreement centers around the homogeneity of soil in the US Corn Belt, in contrast with the huge heterogeneity in China. Our results highlight the importance of soil predictors—especially in China, whose arable areas show considerable heterogeneity. As for individual predictors, the cation exchange capacity of clay (CEC_CLAY) from this group ranked second among all of the predictors, consistent with conclusions from previous research conducted in the North China spring maize zone [53]. In addition, the contribution of soil moisture (SMgs) in this study was surprisingly limited, whereas it brought significant accuracy improvements to the process-based models in previous research [63,86–88]. This is because the soil moisture cannot reflect the physical damage and pollination disturbances caused by excessive rainfall in China. The soil moisture is also likely to become saturated under excessive rainfall; that is, although soil moisture can capture the influence of drought on maize yield, its ability to reflect conditions of excessive wetness is limited.

Remote sensing predictors were the third most important, because they contain unique information about the growing progress and health condition of crops [74]. The land surface temperature did not significantly outperform air temperature in our results, while previous research found that replacing air temperature with land surface temperature (LST) could significantly improve the model accuracy across the US Corn Belt [60]. However, Pede

et al. (2019) only considered climate data and LST, while we used multisource data. Their research assumed that the LST can provide additional canopy information when using only climate data [89]. Our results indicated that this information might have been already captured by vegetation indices, such as the EVI.

Management predictors were the fourth most important, but the nitrogen application total (NAT) ranked third among all predictors. The dramatic increase in grain yield across the Chinese maize belt relied heavily on the application of nitrogen fertilizer [90], which can delay leaf senescence and sustain further grain yield increase—especially for high-yield maize production [91]. Although the application of fertilizer plays an important role in increasing grain yields in China [92–94], it has seldom been considered in yield prediction in previous research. Thus, our results encourage future studies to apply management predictors.

Machine learning models in this study highly outperformed MLR, due to MLR's limitations in dealing with nonlinear relations and collinearity. Machine learning algorithms allow for the extraction of information about complicated interplays of various predictors, and can be performed independently from previously defined interrelationships [82,95]. The XGB model is comparable to the BNN, and similar results were found in previous research conducted in the USA and China [61,96]. This may be because it is hard to determine the optimal structure of a BNN, while the optimal parameters of XGB can be determined by cross-validation or grid search [96]. Furthermore, XGB showed better performance than RF, which is one of the most popular methods in yield prediction [33,53,61,97]. The superiority of XGB can be attributed to its boosting technique, which iteratively reduces bias and variance. Compared with RF, XGB fits each tree on a modified version of the original training dataset, i.e., every new tree uses information from previously grown trees [61].

*4.3. Limitations and Future Work*

This research is not free of uncertainties. First, our results could be significantly improved if we had access to the masks, which could present changes in the planting area from year to year. We only considered planting masks of 2000, 2005, and 2010 in this study due to lacking annual data, largely limiting the performance of our results. This problem will be solved as the cropland masks become increasingly available at high frequency and spatial resolution. Second, the generalization ability of machine learning models can be improved. For example, the XGB model trained at the municipal level succeeded in explaining around 81% of the municipal-level yield variation in the testing process (Figure S6), but only explained around 54% of the county-level yield variation (Figure 4). The weak generalization ability can be ascribed to the domain shift—the models trained from the municipal-level data may lose validity when directly applied to estimate yields at the gridded level, since they have different data distributions [98]. This problem will likely be addressed by integrating the process-based models or transfer learning [99–102].

**5. Conclusions**

This study developed a spatial disaggregation method based on machine learning algorithms and multisource data to produce spatially explicit gridded yields. Compared with the traditional disaggregation method, our method comprehensively considers the information contained in the multisource data, mining the relationships between them and maize yield through machine learning algorithms, while still maintaining strong flexibility and scalability. We found that XGB outperformed MLR and RF, and was comparable to the BNN. The disaggregated maps could reproduce 54% of the county-level yield variability when it was validated by finer-resolution statistical yields. The combination of machine learning and multisource data in producing spatially explicit yields improved upon the existing gridded datasets. Our method is not limited to using MODIS imagery as inputs, and shows promise in generating yield maps at much higher resolution and larger scales. Its accuracy could be further improved by transfer learning and integrating process-based models.

**Supplementary Materials:** The following supporting information can be downloaded at: https://www.mdpi.com/article/10.3390/rs14102340/s1, Figure S1: The spatial distribution of agricultural meteorological stations; Figure S2: Flow chart of MLR; Figure S3: Flow chart of RF; Figure S4: Flow chart of XGB; Figure S5: The architecture of the BNN used in this study; Figure S6: Scatter plots between estimated and statistical yields at the prefecture-scale for the test set; Figure S7: Yield times series in the selected locations for different datasets; Table S1: Default parameters of XGB and RF; Table S2: The RMSE and R2 between different datasets and site records.

**Author Contributions:** S.C.: Data Curation, Methodology, and Writing—Original Draft. W.L. and P.F.: Writing—Review and Editing. T.Y.: Funding Acquisition, Methodology, Resources, and Writing— Review and Editing. Y.M. and Z.Z.: Methodology and Writing—Review and Editing. All authors have read and agreed to the published version of the manuscript.

**Funding:** This research was funded by the National Natural Science Foundation of China grant number NSFC 42171075. The project was supported by the State Key Laboratory of Earth Surface Processes and Resource Ecology. This research was funded by the Program of Introducing Talent to Universities of China (111 Project) grant number BP0820003 and the Strategic Priority Research Program of the Chinese Academy of Sciences grant number XDA28060200.

**Data Availability Statement:** All results and models presented in this study are available on request from the corresponding author for research purposes.

**Conflicts of Interest:** The authors declare that they have no known competing financial interests or personal relationships that could have appeared to influence the work reported in this paper.

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
