# Peer review of "Improving Spatial Disaggregation of Crop Yield by Incorporating Machine Learning with Multisource Data: A Case Study of Chinese Maize Yield"

_remotesensing, doi:10.3390/rs14102340_

Round 1

Reviewer 1 Report

This paper developed a spatial disaggregation method using machine learning and multiple source data to estimate crop yield.

Data and variables, models and methods were well explained and presented.

The results were thoroughly discussed and supported the conclusions.

It would be better if the authors could compare ANNs with the MLR, RF, and XGB models used in this paper.

In addition, please fix citation issues. (Page 4 line 1, Error! Reference source not found.)

Reviewer 2 Report

This paper addresses the yield disaggregation problem. A new method is proposed that leverages machine learning and multi-source data. The approach and results are sound, though I am providing a few suggestions for the authors to consider.

L96: The term "MapSPAMs" is used here as opposed to "MapSPAM" elsewhere. Is there a reason for the difference?

Section 2.2.2: I would love to see a figure of illustration to explain MLR, RF, and XGB, and highlight their similarities and differences.

Figure 6: Is there a reason for the discontinuity in the green line in the Bachu subplot?

Across the document I've got there are several places showing "Error! Reference source not found".

Reviewer 3 Report

Line 50 to 51. Cite typo.

Line 112 to 115. In my opinion, comparison is not needed, arguing why you used MODIS is enough.

Table 1. Links do not work. You may want to cite differently.

Line 194. Which "default settings" do you refer to?

Line 220 to 225: Are “raw grinded maize yield” (line 220), “raw grinded yield” (line 221), “raw estimated yield” (line 225) the same concept?

Line 385 to 390. Discussion of opposite conclusions about water influence on maize yield is made arguing the different conditions of water source for crops and soil heterogeneity. I thought that the solid moisture predictor could capture this situation. Nevertheless, it had lower relative importance at the feature importance analisis. Could you tell me the relative importance of soil moisture that you obtained?
